# Preliminary Results of an Exercise Program After Laparoscopic Resective Colorectal Cancer Surgery in Non-Metastatic Adenocarcinoma: A Pilot Study of a Randomized Control Trial

**DOI:** 10.3390/medicina56020078

**Published:** 2020-02-14

**Authors:** Gabriele Mascherini, Maria Novella Ringressi, Jorge Castizo-Olier, Georgian Badicu, Alfredo Irurtia, Laura Stefani, Giorgio Galanti, Antonio Taddei

**Affiliations:** 1Sport and Exercise Medicine Unit, Dipartimento di Medicina Sperimentale e Clinica, Università degli Studi di Firenze, 50134 Firenze, Italy; gabriele.mascherini@unifi.it (G.M.); laura.stefani@unifi.it (L.S.); giorgio.galanti@unifi.it (G.G.); 2Multidisciplinary Oncology Group, Dipartimento di Medicina Sperimentale e Clinica, Università degli Studi di Firenze, 50134 Firenze, Italy; mnringressi@hotmail.it (M.N.R.); antonio.taddei@unifi.it (A.T.); 3Tecno Campus Mataró-Maresme, EscuelaSuperior de Ciencias de la Salud, 08302 Mataró, Barcelona, Spain; 4Department of Physical Education and Special Motricity, Faculty of Physical Education and Mountain Sports, Transilvania University of Brasov, 500068 Brasov, Romania; 5Departament de la Presidència, InstitutNacional d’EducacióFísica de Catalunya (INEFC), 08038 Barcelona, Spain; airurtia@gencat.cat

**Keywords:** supervised, unsupervised, home-based, physical activity, oncology, cancer survivors, bioimpedance

## Abstract

*Background and objectives:* Performing physical exercise after a colorectal cancer diagnosis is associated with lower mortality related to the tumor itself. In order to improve physical recovery after elective surgery, there are no specific exercise protocols after discharge from the hospital. The purpose of this study is to show the preliminary results of an exercise program after colorectal cancer surgery. *Materials and Methods*: Six patients with non-metastatic colorectal adenocarcinoma addressed to respective laparoscopic were randomly assigned to a mixed supervised/home-based exercise program for six months and compared to a control group without exercise. To assess the effectiveness of the program, functional and body composition parameters were evaluated. *Results:* Three months after surgery, the exercise group increased flexibility (*p* <0.01, ES = 0.33), strength of lower limbs (*p* <0.01, ES = 0.42) and aerobic capacity (*p* <0.01, ES = 0.28). After surgery, the six patients experienced a significant reduction in body mass index (BMI) and free fat mass. More specifically, fat mass reached the lowest values, with a concomitant increase in cell mass after six months (*p* <0.01, ES = 0.33). This did not occur in the control group. *Conclusions:* Colorectal cancer treatment induces a reduction in physical function, particularly during the first six months after treatment. A mixed exercise approach appears promising in countering this process after colorectal cancer surgery.

## 1. Introduction

In 2016, about 30% of deaths in Italy were related to cancer, which is the second leading cause of death after cardio-circulatory diseases (37%). In particular, colorectal cancer is the second highest cause of cancer deaths in all age groups [1].

The survival rate after 10 years from diagnosis is slightly lower than at 5 years, with values of 64% and 58% for colonic and rectal location, respectively; the surgery is the most common treatment for colorectal cancer [2]. Colorectal surgery is often associated with a significant hospital stay (eight days for open surgery and five days for laparoscopic surgery) [3]. 

During the hospital stay for elective colorectal surgery, patients suffer from nausea, pain at rest, impaired intestinal function, and a new wound in the abdomen. This compromises physical function and nutritional status in these patients [4]. In order to enhance recovery after surgery, numerous protocols have been established during the perioperative period until hospital discharge [5,6].

Physical activity after a diagnosis is associated with both lower risk of death from all-causes mortality and colorectal cancer-specific mortality in colorectal cancer survivors [7]. In fact, those who are engaged in physical activity in their leisure time have a risk of death that is 31% lower than that of those who do not, regardless of their physical activity in their leisure time before diagnosis [8].

After six months from colorectal surgery, half of these patients do not return to preoperative functioning and higher levels of physical activity after surgery is associated with the enhanced recovery of physical functioning [9]. Most of the differences in physical activity level after colorectal surgery can be attributed to differences in functioning aspects and experienced symptoms related to cancer and its treatment [10].

To our knowledge, no studies have been designed to determine exact exercise programs for specific cancer types, in particular after colorectal cancer surgery [11]. Therefore, healthcare professionals can only provide generic support about physical activity in colorectal cancer survivors through the interpretation of current guidelines [12]. 

The purpose of this study is to verify the effectiveness of an exercise program, in terms of functional and body composition parameters, after laparoscopic resective colorectal cancer surgery.

## 2. Experimental Section

### 2.1. Study Population

Eight subjects were enrolled for the preliminary results shown in this study. One declined participation after the preoperative evaluation because adherence to the following control evaluations was not guaranteed. One was excluded from the second evaluation because their histological diagnosis did not fall within the inclusion criteria of the study. Therefore, the sample was represented by six patients (four males and two females, aged 72.9 ± 7.3 years, height 168.5 ± 7.1 cm) with non-metastatic colorectal adenocarcinoma (Stages I and II) addressed to laparoscopic resective intervention.

Inclusion criteria: histological diagnosis of adenocarcinoma, eligibility for laparoscopic colonic surgery (right hemicolectomy, intermediate colectomy, anterior rectal resection), life expectancy at least 6 months/1 year, no previous similar surgical intervention, age > 18 years, patients living in the city of the study or in surrounding areas.

Exclusion criteria: participation in other research projects potentially capable of altering the results of this study, absence of informed consent, mental or psychiatric pathological conditions that interfere with the protocol, relative and absolute contraindications related to physical exercise, previous or scheduled radio or chemotherapy, primary or secondary immunodeficiency or presence of autoimmune diseases, travel in exotic areas in the last five years, and antibiotic treatment in the six months prior to surgery.

Subjects were enrolled after receiving written informed consent. The study was carried out in conformity with the ethical standards laid down in the 1975 Declaration of Helsinki and was approved by the Ethics Committee for Clinical Trials of the Area Vasta Centro of the Tuscany Region in Florence (code OSS.16.237 with date of approval 22 July 2016).

### 2.2. Preoperative Workup and Surgical Techniques

Diagnosis was determined by a pancolonoscopy with multiple biopsies. In cases of incomplete colonoscopy, computed tomographic (CT) colography was performed. The pretreatment tumor stage was determined in all patients by chest and abdominal CT scans.

All the surgical procedures were performed by laparoscopic access at the General Surgical Unit of Careggi University Hospital in Florence. Elective laparoscopic colon resections were performed through a standardized medial-to-lateral approach, with proximal ligation of vascular pedicles [13].

Tumors were staged according to the current American Joint Commission on Cancer/International Union Against Cancer TNM staging system [14].

### 2.3. Study Design

Randomized controlled clinical trial.

Eligible patients were randomized into two groups:

(1) Control group, these patients were subjected to the evaluations with standard care indications for physical activity.

(2) Exercise group, in addition to the evaluations these patients had a personalized exercise program.

The Multidisciplinary Oncology Group of tumors of the lower digestive tract enrolled patients. After the randomization and before surgery, patients underwent a first evaluation of functional and body composition parameters at the Sports Medicine Center.

The scheduled frequency of evaluations was (T1) Pre-surgery, day of hospitalization; (T2) Post surgery, day of hospital discharge; (T3) 30 days from discharge; (T4) 90–100 days from discharge; (T5) 6 months from discharge (Figure 1).

### 2.4. Functional Assessment

#### 2.4.1. Cardiovascular Fitness

The 6-Minute Walking Test (6-MWT) is a simple practical test that requires a 100 ft hallway, but no exercise equipment or advanced training for technicians. This test measures the distance that a patient can quickly walk on a flat, hard surface in a period of 6 min. However, because most activities of daily living are performed at sub maximal levels of exertion, this test may better reflect the functional exercise level for daily physical activities, and evidence supports the 6-MWT as a valid measure of recovery after colorectal surgery [15]. The parameters recorded during 6-MWT were: distance covered (6-Minute Walking Distance (6-MWD)), peak heart rate with heart rate monitor, systolic and diastolic blood pressure at rest and at the end of the test, and self-perception of effort (with CR10 scale) [16].

#### 2.4.2. Muscular Fitness

Flexibility and muscle strength evaluations were performed with easily executable and reproducible tests in an outpatient setting as a sit and reach test for flexibility [17], a hand grip test to estimate the overall static strength of the upper limbs [18], and a chair test was also used, to assess the strength of the lower limbs [19].

### 2.5. Body Composition Analysis

Body composition is a recognized method for assessing nutritional status and is a parameter of health-related physical fitness. 

#### 2.5.1. Anthropometry

An anthropometric assessment was performed following the protocols and standards of the International Society for the Advancement of Kinanthropometry, ISAK [20], by the same operator with a wide experience. Body mass (BM) was measured to the nearest 0.1 kg and stature (h) to the nearest 0.5 cm (Seca 700^®^ with stadiometer, Seca Corp©, Hamburg, Germany). Body mass index (BMI) was calculated as BM divided by h squared (kg/m2). Circumference measures were made with an anthropometric tape (Holtain 1.5 m flexible tape, Holtain Limited^®^, Crymych, United Kingdom) at waist, hip and arm relaxed, in cm. The triceps, subscapular, biceps and supraspinal skinfolds were measured, in mm (Holtain Tanner/Whitehouse skinfold caliper 610 ND, Holtain Limited^®^, Crymych, United Kingdom). The sum of the four skinfolds was calculated (mm). 

#### 2.5.2. Whole-Body Bioimpedance Analysis (BIA) and Vector Analysis (BIVA)

Whole-body impedance (Z) is the body tissues’ opposition to the flow of an electric current and it is the vector sum of the resistance (R, Ω)—the major resistance to the current through intracellular and extracellular ionic fluids—and the reactance (Xc, Ω)—the additional opposition due to the capacitive elements such as cell membranes, tissue interfaces, and non-ionic substances. The present study measured these parameters using a phase-sensitive, previously calibrated device (BIA 101 Sport Edition, Akern, Florence, Italy) that emits an alternating sinusoidal electric current of 400 mA at an operating single frequency of 50 kHz. The measurements were performed using low intrinsic impedance electrodes (Biatrodes, Akern, Florence, Italy), through the standard whole-body, tetrapolar, distal BIA technique [21]. From raw impedance variables (i.e., R, Xc and PA), other parameters were derived: total body water (TBW), extracellular water (ECW), intracellular water (ICW), body cell mass (BCM), fat-free mass (FFM), and resting metabolic rate (RMR). In order to assess the body hydration and BCM independently of regression equations, the Bioelectrical Impedance Vector Analysis (BIVA) was used. BIVA uses the raw impedance parameters, standardized by height in order to remove the effect of conductor length, yielding a vector, which is plotted in an RXc graph [22]. The length of the vector indicates hydration status from fluid overload (decreased resistance, short vector) to exsiccosis (increased resistance, longer vector), and a sideways migration of the vector due to low or high reactance indicates a decrease or increase in the dielectric mass (membranes and tissue interfaces) of soft tissues [23]. The individual vector can be ranked on the RXc point graph with regards to tolerance ellipses representing 50%, 75% and 95%, according to the values of a reference population [22].

### 2.6. Exercise Program

The exercise consisted of a combined aerobic and resistance training program [24]. The aerobic exercises were individually set in terms of heart rate and perceived effort based on the 6-MWT performed on the day of hospital discharge and updated according to the follow-up. Walking was the activity prescribed and this was carried out unsupervised at least 3 times a week for 30 min each session.

The resistance exercise was scheduled twice a week with the supervision of an exercise physiologist for the first 30 days after hospital discharge. The program consisted of eight exercises involving different muscle groups with 3 sets and 8–12 repetitions. After learning the execution, the resistance exercise program was continued in an unsupervised way and updated according to the follow-up [25].

### 2.7. Statistical Analysis

Means and standard deviations were calculated for all variables, and the normality of the sample distributions was confirmed with the Shapiro–Wilk test. Repeated measures of Analysis of Variance (ANOVA) were applied between the different checkpoints of the study (T1 to T5). A paired Student’s t-test was used to establish the differences between T1 and every checkpoint. The Cohen’s f2 Effect Size (ES) was calculated to determine the magnitude of effect, by standardizing the coefficients according to the appropriate between-subjects standard deviation. ES was assessed using the following criteria: small < 0.02, medium < 0.15, and large < 0.35 [26]. P value was adjusted by Bonferroni correction (0.05/4 comparisons), considering *p* adjusted = 0.01 as the statistically significant value. The data were analyzed using the IBM SPSS 16.0 Statistics for Windows statistical package (SPSS Inc., Chicago, IL, USA). Specifically, the whole-body bioimpedance vectors were analyzed by the RXc graph method using the BIVA software [27]. Each individual was plotted in the tolerance ellipses (50%, 75% and 95%) of their healthy male and female reference populations [28], and BIA vector migration from T1 to T5 of both groups (exercise and control groups) was plotted.

## 3. Results

The study sample underwent the five assessments scheduled within the timeline of the study design. The three patients in the exercise group performed all eight scheduled supervised training sessions. Compliance with unsupervised aerobic activity was verified and encouraged during supervised sessions in the first month after hospital discharge, and subsequently during control assessments.

### 3.1. Functional Assessment

All the functional parameters decrease their performance just after surgery (T1 vs. T2), however none of them do so significantly, either in the physical exercise group, or in the control group (Table 1). Statistically significant differences (*p* < 0.01) from post-surgery (T2) were attained at T4 (3-month post-surgery) and T5 (6-month post-surgery), when the group that continued to carry out the program of unsupervised physical exercise began to obtain plausible benefits on flexibility (sit and reach: T1 = −15.3 ± 9.2, T2 = −17.3 ± 11.0, T5 = −9.9 ± 5.5 cm, *p* < 0.01, ES = 0.33), strength of lower limbs (30″ Chair test: T1 = 11.3 ± 4.5, T2 = 10.0 ± 5.6, T5 = 14.1 ± 5.1 rep, *p* < 0.01, ES = 0.42) and aerobic capacity (6-MWD: T1 = 463.3 ± 102.6, T2 = 418.3 ± 107.5, T5 = 472.3 ± 105.1 m, *p* < 0.01, ES = 0.28).

### 3.2. Body Composition Analysis

After surgery (T1 vs. T2), the six patients experienced a significant reduction in body mass and BMI, associated with a significant loss of FFM (Table 2), ICW and BCM (Table 3). According to the BIVA, a concordant vector migration to the right was observed after surgery (Figure 2). In the exercise group that underwent the mixed exercise program, at the end of the six-month study (T5), both body mass and BMI registered the lowest values, associated with a statistically significant decrease in hip circumference (Table 2), and an increase in ICW, BCM and RMR (Table 3). This recovery process, especially referring to the values linked to the FFM (ICW, BCM, RMR), to the baseline values prior to surgery, were detected by the BIVA (Figure 2), which at the end of the study tends to migrate towards its initial values (T1 vs. T5). This approximation of the BIVA is common for both groups after six months, although the bioelectrical migratory kinetics vary between both groups previously, from T3 to T4 and T4 to T5 (Figure 2). 

After surgery, the continuous and statistically significant increase in body mass in the control group, and, on the contrary, the optimization of body composition in the exercise group due to the mixed exercise program, could explain this differentiated bioelectrical behavior. In any case, it should be noted that at the end of the study, there were no significant increases in FFM induced by the exercise program applied (Table 2).

Figure 2 shows the BIVA follow-up in detail. The mean vector of both groups experienced a shift to the right after the surgery. The migration along the different time points from post-surgery to six months after discharge differed between groups. Thirty days after discharge, both groups showed vector displacements to a downwards-left position, maybe due to an increase in body fluids. After this point, the vector paths of both groups from T3 to T4 were opposite: while the control group continued the previous trend, the exercise group showed an upwards migration, due to a greater Xc (Figure 2A). The individual vector displacement from T1 to T5 is shown in Figure 2B. The subjects in the exercise group (S1, S2, S3) all show downward migration, while in the control group (S4, S5, S6) there is no homogeneous migration. 

The BIVA assessment shown in Figure 3 also demonstrates that five out of six patients were in the cachexic quadrant before surgery.

## 4. Discussion

The importance of physical exercise in cancer survivors is well established [25]. However, the different studies that address this issue should be individually analyzed considering the characteristics of the subjects enrolled and the exercise program. Intervention studies with larger samples generally include patients with non-uniform treatments.

Colorectal cancer treatment contributes to the deterioration of the functional status, particularly during the first six months of treatment [29]. An intervention study performed by Devin et al. has included patients undergoing several treatment schemes such as surgery only, surgery and chemotherapy, surgery and radiotherapy, surgery and chemotherapy and radiation, radiation and chemotherapy [30]. This make the interpretation of the results and the criticalities that could specifically occur in cancer survivors difficult. Our study established restricted inclusion criteria makes the sample homogeneous, although non-extended. Moreover, each patient may have specific rehabilitation needs related to the surgery and the cancer itself. The specific post-operative pain and numbness can affect both the ability and the desire to exercise in previously enjoyable ways. Therefore, in the early survivorship stage an exercise physiologist specialist should be designated for these patients.

The characteristics of the subjects also influence the program of physical exercise that can be proposed, since the optimal type and frequency of exercise that will mostly enhance recovery in the early survivorship stage of cancer is until today unknown [25].

The supervised exercise proved effective within a 12-week program, but we have no information on whether the subjects continued to train after the end of the study protocol [31]. On the other hand, unsupervised exercise moved to provide less improvements but physical activity levels remain as the guidelines recommend even six months from the start of the study [32].

In a pilot RCT [33], no results were obtained with an early exercise program performed in a hospital stay in stage 1~3 colorectal cancer patients. Another study with a large sample size reported that supervised rehabilitation programs were more likely to have functional capacity go back to baseline values at four weeks after surgery, in comparison with unsupervised programs [34]. Gillis et al. achieved greater functional capacity improvement with a pre-habilitation exercise program in patients scheduled for elective colorectal cancer surgery in comparison with the same program started after surgery [35]. However, the exercise program in this study was followed exclusively in an unsupervised way.

The “supervised/unsupervised” mixed model proposed in this study allows for monitoring and updating of the training program. This design was decided because the training program also occurs during a critical period, like the first month after discharge from the hospital. In fact, the patients have to come back to the hospital twice a week to perform a supervised resistance exercise and update the unsupervised aerobic exercise.

The exercise program of the present study was planned according to the American College of Sports Medicine guidelines [36]. These may appear non-specific, but if carried out under the supervision of a specialist, a personalized intervention can be performed modulating the application.

The results of the present study should be considered as preliminary. However, they are promising since the initial improvements of the exercise group have been obtained through a sustainable modality, without an excessive process of medicalization of physical exercise.

The distance covered during the 6-MWT, a valid post-surgical recovery measure [15], had already increased in the exercise group in the first month after discharge, while the control group did not reach baseline values even at six months from the start of the study. Body composition shows how the exercise program was effective. This is demonstrated by the gradual reduction of fat mass with a simultaneous increase in cell mass.

The BIVA follow-up reports a shift to the right after the surgery in both groups, possibly reflecting a reduction in body fluids and BCM (defined as metabolically active tissues in the body including muscle cells, organ cells, blood cells and immune cells) [37]. The displacements of the vectors that occur in both groups in T3 may be due to an increase in body fluids, while the different behaviors from T3 to T4 would reflect an increase in both body fluids and BCM. Despite the fact that after T3 the exercise program was not supervised and that both groups returned to vector values close to the baseline from T4 to T5, the different vector behavior from T3 to T4 could reflect the beneficial effect of following an exercise program, which could allow a faster recovery compared to the control group.

The individual assessment of BIVA migration shows that the exercise group has a similar trend in all three subjects. At the end of the study, the subjects who followed the training program were placed lower in the RXc graph than the position recorded at the baseline. This could be due to an increase in body water, due to a concomitant increase in RZ and a decrease in XC over time. As for the control group, the vector migration from T1 to T5 shows three different behaviors. In addition, the displacements appear more pronounced than in the exercise group. Therefore, is possible to speculate that exercise is able to transmit changes in body composition in a more homogeneous way, even in subjects with different anthropometric profiles.

In addition, this study provides the characterization and follow-up of the whole-body bioimpedance vectors in colorectal cancer patients for the first time. Despite the fact that these patients would be mainly classified as overweight and obese through other methodologies, five of them were identified in the cachexic quadrant of their reference populations and one was in the limit between the obese and cachexic quadrants. This could be due to a sarcopenic obesity status characterized by increased fat mass and decreased lean mass because of the disease [38]. A correctly identified nutritional status could help in the exercise and nutritional prescription and follow-up. However, ‘classic’ BIVA has a limited sensitivity in assessing the features of body composition (i.e., FM and FFM) due to the lack of consideration of the effect of cross-sectional areas of the body, which interferes with bioelectrical values [39,40]. A promising alternative is ‘specific’ BIVA, a method which proposes the correction of bioelectrical values for body geometry and could help to overcome the mentioned limitation in patients with cancer [41].

The present study has some limitations. First, the two samples had, to some extent, different anthropometric parameters at the beginning of the study. This is due to the very selective inclusion criteria of the study, and furthermore, the subsequent randomization does not allow the creation of homogeneous groups for these parameters. Only an increase in sample size will allow this problem to be solved. Moreover, the small sample size does not allow for generalization of the results obtained, nor can it allow for establishment of the effectiveness of the treatment or the sustainability of the proposed therapeutic path. However, the small sample size does make it possible to trace a direction that allows the hypothesizing of an integration between multiple professionals in the treatment of colorectal cancer.

The strength of this study is the homogeneity of the histologic and therapeutic characteristics of the sample. In fact, there are no chemotherapeutic and/or radiotherapeutic influences that can alter the parameters of physical function or body composition.

## 5. Conclusions

A mixed approach involving supervised and unsupervised exercise appears promising after colorectal cancer surgery. Our proposal, based on clinical evaluations, could provide a faster recovery of physical function. Further studies are needed to confirm these preliminary results.

## Figures and Tables

**Figure 1 medicina-56-00078-f001:**
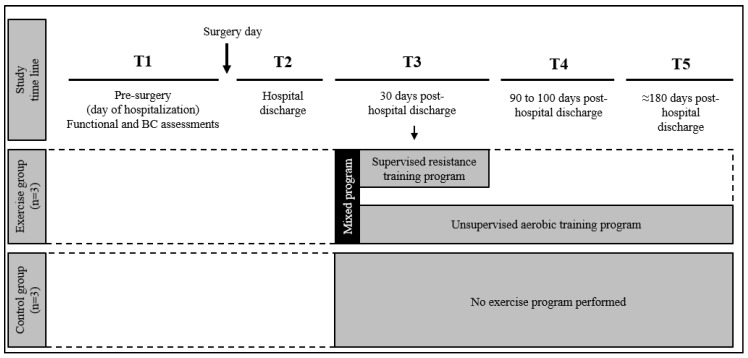
Timeline of the study design.

**Figure 2 medicina-56-00078-f002:**
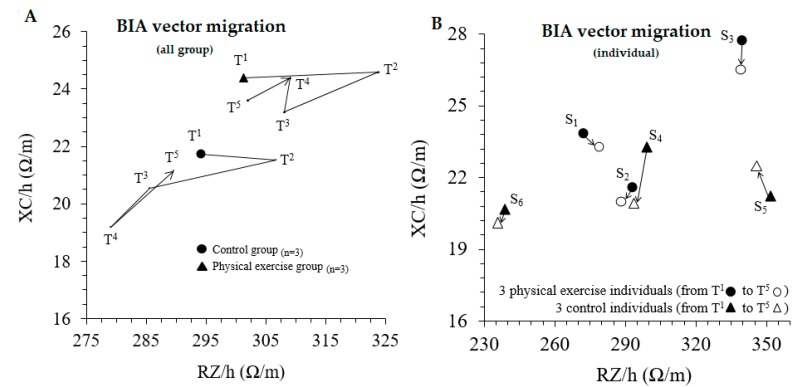
Migration over a period of six months. (**A**) shows the vectors migration of the exercise and control groups from T1 to T5 checkpoints of the follow-up. (**B**) shows the individual vector displacements of the six patients from T1 to T5 RZ/h, height-adjusted resistance; XC/h, height-adjusted reactance.

**Figure 3 medicina-56-00078-f003:**
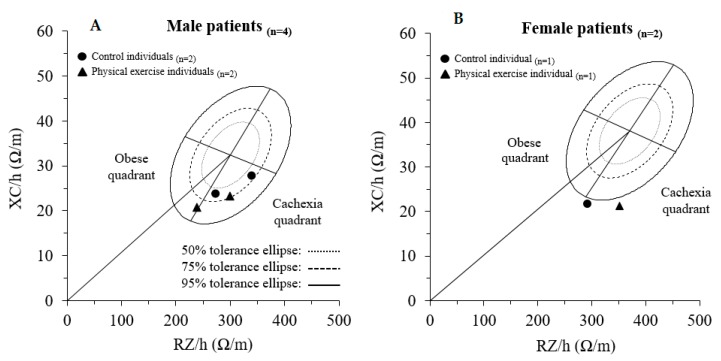
Pre-surgery individual BIVA, according to the corresponding healthy reference population. (**A**) shows the vectors of male patients and (**B**), the vectors of female patients. RZ/h, height-adjusted resistance; XC/h, height-adjusted reactance.

**Table 1 medicina-56-00078-t001:** Functional and cardiovascular variables throughout the study.

	Sit and Reach (cm)	30˝ Chair Test (rep.)	Handgrip Dx. (kg)	Handgrip Sx. (kg)	6-MWD (m)	CR10 (a. u)	Rest HR (bpm)	HR Max (bpm)	Rest SBP (mmHg)	Rest DBP (mmHg)
*Exercise Group*										
T^1^_pre-surgery_	−15.3 ± 9.2	11.3 ± 4.5	29.0 ± 4.6	30.7 ± 6.8	463.3 ± 102.6	4.3 ± 2.5	91.3 ± 4.9	107.3 ± 1.5	118.3 ± 17.6	73.3 ± 10.4
T^2^_post-surgery_	−17.3 ± 11.0	10.0 ± 5.6	26.3 ± 6.7	26.3 ± 6.7	418.3 ± 107.5	4.7 ± 3.1	89.3 ± 5.1	106.7 ± 2.9	106.7 ± 5.8	66.7 ± 2.9
T^3^_post-exercise_	−14.0 ± 7.8	12.7 ± 5.0 § _(0.27)_	29.3 ± 4.6	29.0 ± 6.1	476.7 ± 121.0	5.0 ± 3.0	91.7 ± 4.2	108.0 ± 11.8	110.0 ± 5.0	71.7 ± 2.9
T^4^_3-months_	−12.0 ± 6.0	13.0 ± 6.1 _(0.31)_	27.0 ± 7.8	30.7 ± 4.2	478.3 ± 101.0 § _(0.32)_	5.0 ± 4.0	89.3 ± 6.0	107.0 ± 10.8	118.3 ± 7.6	73.3 ± 2.9
T^5^_6-months_	−9.9 ± 5.5 § _(0.33)_	14.1 ± 5.1 § _(0.42)_	29.5 ± 4.1	28.3 ± 6.0	472.3 ± 105.1 § _(0.28)_	5.6 ± 1.7	89.0 ± 8.0	108.2 ± 2.4	109.3 ± 12.9	70.2 ± 5.0
*Control Group*										
T^1^_pre-surgery_	−8.0 ± 8.9	15.0 ± 5.2	32.7 ± 18.2	33.7 ± 11.7	543.3 ± 102.1	4.7 ± 2.5	94.3 ± 14.6	120.7 ± 10.6	131.7 ± 7.6	76.7 ± 7.6
T^2^_post-surgery_	−7.7 ± 7.2	11.7 ± 4.0	30.7 ± 15.0	30.7 ± 15.0	470.0 ± 87.2	3.3 ± 1.5	90.7 ± 21.1	108.7 ± 17.7	120.0 ± 18.0	66.7 ± 15.3
T^3^_post-exercise_	−9.3 ± 5.9	14.0 ± 5.2	33.3 ± 14.5	30.7 ± 15.0	502.0 ± 88.9	2.8 ± 1.4	92.3 ± 18.6	108.0 ± 9.6	128.3 ± 5.8	73.3 ± 2.9
T^4^_3-months_	−9.3 ± 11.2	15.3 ± 6.8	32.3 ± 13.6	31.3 ± 14.5	506.7 ± 107.9	3.7 ± 1.5	84.3 ± 25.0	102.7 ± 10.8	128.3 ± 5.8	73.3 ± 2.9
T^5^_6-months_	−11.0 ± 9.5	15.3 ± 5.9	33.3 ± 14.5	31.7 ± 14.2	510.0 ± 87.2	3.7 ± 1.5	89.3 ± 18.3	109.7 ± 5.5	121.7 ± 7.6	75.7 ± 2.9

T: checkpoint; 6-MWD: 6-Minute Walking Distance; CR10: Borg scale, with a. u (arbitrary units); HR: heart rate; SBP: systolic blood pressure; DBP: diastolic blood pressure. *p* < 0.016: § (checkpoint _vs._ post-surgery). In brackets; Cohen’s effect size.

**Table 2 medicina-56-00078-t002:** Differences in anthropometric and body composition variables throughout the study.

	Weight (kg)	BMI (kg/m2)	Waist Circ. (cm)	Hip Circ. (cm)	Biceps Circ. (cm)	∑4 Skinfold (mm)	FM (%)	FFM (kg)
*Exercise Group*								
T^1^_pre-surgery_	83.6 ± 19.6	30.1 ± 7.3	103.2 ± 18.5	109.0 ± 17.3	31.5 ± 4.5	129.1 ± 8.1	30.8 ± 10.6	54.7 ± 8.5
T^2^_post-surgery_	80.0 ± 18.6 * _(0.11)_	28.8 ± 6.9 * _(0.10)_	101.7 ± 18.1	106.7 ± 16.9 * _(0.08)_	30.7 ± 4.5	128.1 ± 10.1	30.3 ± 9.9	52.3 ± 8.5 *_(0.16)_
T^3^_post-exercise_	81.3 ± 17.8	29.3 ± 6.7	102.2 ± 17.5	105.7 ± 17.0	30.4 ± 4.1	127.3 ± 8.8	29.9 ± 9.9	53.7 ± 8.0
T^4^_3-months_	81.7 ± 17.8	29.5 ± 6.8	101.2 ± 18.0	102.7 ± 13.1	30.5 ± 4.5	127.1 ± 4.7	29.8 ± 10.4	53.8 ± 7.8
T^5^_6-months_	82.0 ± 18.6	29.6 ± 7.1	100.4 ± 17.2	104.9 ± 16.1	30.5 ± 4.3	125.5 ± 2.8	29.5 ± 9.8	54.2 ± 6.0
*Control Group*								
T^1^_pre-surgery_	71.3 ± 15.2	24.3 ± 2.7	88.0 ± 13.9	101.3 ± 6.0	27.7 ± 2.1	117.3 ± 20.1	25.2 ± 5.1	54.3 ± 13.0
T^2^_post-surgery_	68.6 ± 15.4 * _(0.10)_	23.4 ± 3.0 * _(0.02)_	86.7 ± 14.5	99.7 ± 8.0	26.5 ± 3.0	117.6 ± 19.8	25.3 ± 2.8	52.5 ± 13.0
T^3^_post-exercise_	70.5 ± 15.5 § _(0.07)_	24.1 ± 2.9 § _(0.13)_	87.5 ± 13.9	100.7 ± 6.7	27.0 ± 3.5	117.1 ± 17.4	25.1 ± 4.3	54.3 ± 13.8
T^4^_3-months_	70.8 ± 15.2 § _(0.15)_	24.8 ± 2.8 § _(0.27)_	88.5 ± 15.0	99.3 ± 5.0	27.2 ± 2.5	119.3 ± 16.1	26.1 ± 4.0	54.1 ± 12.9
T^5^_6-months_	71.4 ± 13.1	24.4 ± 1.8	86.5 ± 12.1	99.0 ± 5.0	27.7 ± 2.6	115.3 ± 13.3	27.7 ± 4.8	54.2 ± 12.5

T: checkpoint; BMI: body mass index; FM: fat mass. *p* < 0.016: * (checkpoint _vs._ pre-surgery); § (checkpoint _vs._ post-surgery). In brackets; Cohen’s effect size.

**Table 3 medicina-56-00078-t003:** Differences in bioelectrical parameters, body water compartments, body cell mass and rest metabolic rate throughout the study.

	RZ (Ω)	XC (Ω)	PA (°)	TBW (L)	TBW (%)	ECW (%)	ICW (%)	BCM (kg)	RMR (kcal)
*Exercise Group*									
T^1^_pre-surgery_	502.3 ± 53.5	40.7 ± 5.5	4.6 ± 0.4	41.1 ± 5.9	49.7 ± 7.5	53.1 ± 2.5	46.9 ± 2.5	25.2±5.1	1480.8 ± 149.1
T^2^_post-surgery_	539.7 ± 69.2	41.0 ± 5.6	4.3 ± 0.4 *_(0.42)_	39.2 ± 6.0 * _(0.18)_	48.6 ± 6.4	54.8 ± 2.3 * _(0.40)_	45.2 ± 2.3 * _(0.40)_	23.2±4.9 * _(0.23)_	1421.5 ± 141.3 * _(0.23)_
T^3^_post-exercise_	513.7 ± 42.7	38.7 ± 2.3	4.3 ± 0.4	40.3 ± 5.5	49.8 ± 7.9	55.0 ± 2.7	45.0 ± 2.7	23.7±5.1 *	1437.1 ± 148.8
T^4^_3-months_	515.3 ± 36.6	40.7 ± 1.2	4.5 ± 0.2	39.4 ± 5.8	53.3 ± 3.3	53.7 ± 1.3	46.3 ± 1.3 § _(0.32)_	24.2±4.2 § _(0.31)_	1451.5 ± 121.8 § _(0.18)_
T^5^_6-months_	515.0 ± 48.0	40.3 ± 5.0	4.5 ± 0.2	40.1 ± 4.2	49.4 ± 6.9	53.7 ± 1.1	46.3 ± 1.1 § _(0.33)_	24.5±2.9 § _(0.33)_	1461.2 ± 84.0 § _(0.26)_
*Control Group*									
T^1^_pre-surgery_	501.0 ± 68.4	37.0 ± 3.0	4.3 ± 0.8	41.4 ± 9.5	58.1 ± 2.6	55.5 ± 5.2	44.5 ± 5.2	24.0 ± 8.4	1446.0 ± 244.2
T^2^_post-surgery_	522.3 ± 93.9	36.7 ± 6.1	4.1 ± 1.0 * _(0.13)_	40.2 ± 9.7	58.5 ± 1.4	57.1 ± 6.7	42.9 ± 6.7 * _(0.17)_	22.4 ± 9.1 * _(0.11)_	1399.6 ± 262.4
T^3^_post-exercise_	486.3 ± 87.3 § _(0.22)_	35.0 ± 3.0	4.2 ± 0.9	42.3 ± 10.4	59.8 ± 2.0	57.1 ± 5.9	42.9 ± 5.9	22.7 ± 9.2	1419.0 ± 267.5
T^4^_3-months_	475.3 ± 85.5 § _(0.29)_	32.7 ± 3.2	4.2 ± 0.7	43.2 ± 10.4 § _(0.11)_	59.2 ± 2.1	57.5 ± 4.8	42.5 ± 4.8	22.7 ± 7.8	1409.2 ± 226.0
T^5^_6-months_	491.0 ± 70.1	36.3 ± 0.6	4.3 ± 0.7	42.9 ± 9.6	59.1 ± 2.9	57.4 ± 4.6	42.6 ± 4.6	23.1 ± 8.1	1419.4 ± 235.5

RZ: resistance; XC: reactance; PA: phase angle; TBW: total body water; ECW: extracellular water; ICW: intracellular water, BCM: body cell mass; RMR: rest metabolic rate. *p* < 0.016: * (checkpoint _vs._ pre-surgery); § (checkpoint _vs._ post-surgery). In brackets; Cohen’s effect size.

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
