# Peer review of "Preliminary Results of an Exercise Program After Laparoscopic Resective Colorectal Cancer Surgery in Non-Metastatic Adenocarcinoma: A Pilot Study of a Randomized Control Trial"

_medicina, 2020, doi:10.3390/medicina56020078_

Round 1
Reviewer 1 Report
Thank you for submitting the manuscript titled: “Effectiveness of an exercise program after laparoscopic resective colorectal cancer surgery in non-metastatic adenocarcinoma: a pilot study of a randomized control trial” for review.
Please find my reviewers comments as constructive for subsequent improvement in the manuscript.
I think this is novel in that you have considered a specific group of patients with specific cancer, and offer a PA programme and follow-up. So its novel in that it focuses on PA for those post colorectal cancer surgery, but only in the sense you offer a PA programme, not that the PA programme itself has been designed with a theoretical concept in mind or other element of novelty? or can you clarify your study/intervention design with reference to a behaviour change theory- or other?
However you need to tidy up the writing style and follow a systematic layout. For example, you have presented additional results in the discussion, which should be in the results section. It’s not the easiest read.
Title- add preliminary. Preliminary-Effectiveness of an exercise program…
Abstract
Line 28 “three months after surgery, the exercise group”
Line 31 “ specifically, fat mass reached 30 the lowest values, with a concomitant increase in cell mass after 6 months. This did not occur in the control group.” was it significant for the exercise group?
Check readability line 33.” Colorectal cancer treatment induces a worsening of the physical condition”
Check readility line 42 “colorectal cancer stands at the place leading cause of cancer deaths in all age groups
Introduction
Line 61, “ so far no studies “…. To our knowledge, no studies….
Please consider providing more detail from the literature such as the following more detailed summary of PA recommendations : Brigid M. Lynch, Eline H. van Roekel & Jeff K. Vallance (2016) Physical activity and quality of life after colorectal cancer: overview of evidence and future directions, Expert Review of Quality of Life in Cancer Care, 1:1, 9-23, DOI: 10.1080/23809000.2016.1129902 … “Cancer survivors are advised to avoid inactivity and return to normal daily activities as soon as possible; to engage in moderate-to-vigorous physical activity (MVPA; activities that expend ≥3 metabolic equivalents [METs]) [24] at least 150 min/week; and, include strength training on at least two occasions per week. If older age or comorbid chronic conditions limit cancer survivors’ ability to engage in physical activity, it is recommended that survivors are as active as their abilities allow, and that they avoid long periods of inactivity.[22,23] These recommendations were generated in response to epidemiological research demonstrating that physical activity reduces colorectal cancer recurrence and mortality in a dose–response fashion.[25]”
Participants- study population, 6 from a sample of how many eligible? Did any decline participation? Did any initially consent but then drop out of the exercise trial? How were participants actually recruited/invited to participate in this study?
How were the 6 patients randomised?
Participants on the control group. Line 99 suggests ‘no specific indication for PA’ so here you have gone against clinical guidelines and not referenced any PA recommendation, or did you offer ‘standard care’ in which patients were informed PA was recommended- but not given any specific intervention/support for PA?
161- the exercise were individually set in terms of heart rate…… How were they set, in what way, what exercises were set? How has the personalisation occurred. Please provide specific detail.
Basically at T3 all patients make some improvement following discharge. Between T3 &4 the exercise group perform better, given they are encouraged to engage in behavioural activity self-directed after receiving a structure and supervised exercise training period. The control group do less well, and are returning to T1 data, by T5 all patients are returning to T1.
Please consider the very limited sample with reference to the statistics, and refer to possible type 1/ 2 errors. Based on the pilot data what sample size would be required to run a full-RCT (please present a sample size calculation considering the power and effect sizes)
Is this more about reporting the feasibility and acceptability of a proof of concept study- rather than the preliminary effectiveness given the exercise group was n3? The next stage would be to consider the effectiveness via a scaled RCT.
minor
Line 88- when you abbreviate please ensure that the words have capitals (check throughout). For example Computed Tomographic (CT)… BMI / BIA/ BIVA etc…
Please be consistent with language so in some parts you refer to the ‘exercise group’ and in other parts you refer to the exercise for ‘experimental group’.
Author Response
The authors would like to thank to reviewers for their precious and constructive advice. We found the comments very helpful in improving the paper.
We have attached the responses point-by-point.

Reviewer 2 Report
This is an interesting pilot study, but it must be improved for the further consideration.
The main problem of this paper is emphasizing the effectiveness in a very very small sample of participants (n= 3 in each group) and instead, not emphasizing the limitations of the study. The other problem is data analysis of the results with standard deviations in a small sample.
Title: The title should be made different, without "effectiveness".
Keywords: Why so many keywords? please reduce it.
Study population: the gender is missing
Results: I suggest the individual data analysis for each patients alone in new tables since there are only 6 of them, to get a better insight in all changes. Additionally, please discus it in the discussion part.
Compliance is missing and the participants' flow diagram.
Limitation part of discussion should be better written with more explanations.
Conclusion part: From this small sample study, exercise cannot be concluded as effective. Such pilot RCT research can provide preliminary proof-of-concept data and help develop effective interventions.
Author Response
The authors would like to thank to reviewers for their precious and constructive advice. We found very helpful in improving the paper.
We attach the responses point-by-point.

Round 2
Reviewer 1 Report
Der Authors. Thank you for systematically addressing my comments and feedback. I can see that you have made significant improvements to the paper. Kindest regards
Reviewer 2 Report
The authors have improved their manuscript.